# OLIVES Dataset: Ophthalmic Labels for Investigating Visual Eye Semantics

**Mohit Prabhushankar**[1], **Kiran Kokilepersaud**[1]*, **Yash-yee Logan**[1]*,
**Stephanie Trejo Corona**[2]*, **Ghassan AlRegib**[1], and **Charles Wykoff**[2]

[1]OLIVES at the Centre for Signal and Info. Processing, Georgia Tech, Atlanta, GA 30332, USA
[2]Retina Consultants Texas, Retina Consultants of America, Houston, Texas 77030, USA
`{mohit.p, kpk6, ylogan3, alregib}@gatech.edu`,
`{stephanie.trejo, ccwmd}@retinaconsultantstexas.com`

## Abstract

Clinical diagnosis of the eye is performed over multifarious data modalities including scalar clinical labels, vectorized biomarkers, two-dimensional fundus images, and three-dimensional Optical Coherence Tomography (OCT) scans. Clinical practitioners use all available data modalities for diagnosing and treating eye diseases like Diabetic Retinopathy (DR) or Diabetic Macular Edema (DME). Enabling usage of machine learning algorithms within the ophthalmic medical domain requires research into the relationships and interactions between all relevant data over a treatment period. Existing datasets are limited in that they neither provide data nor consider the explicit relationship modeling between the data modalities. In this paper, we introduce the Ophthalmic Labels for Investigating Visual Eye Semantics (OLIVES) dataset that addresses the above limitation. This is the first OCT and near-IR fundus dataset that includes clinical labels, biomarker labels, disease labels, and time-series patient treatment information from associated clinical trials. The dataset consists of $1268$ near-IR fundus images each with at least $49$ OCT scans, and $16$ biomarkers, along with $4$ clinical labels and a disease diagnosis of DR or DME. In total, there are 96 eyes' data averaged over a period of at least two years with each eye treated for an average of 66 weeks and 7 injections. We benchmark the utility of `OLIVES` dataset for ophthalmic data as well as provide benchmarks and concrete research directions for core and emerging machine learning paradigms within medical image analysis.

## 1  Introduction

Ophthalmology refers to the branch of medical science that deals with the structure, functions, diseases, and treatments of the eye. A stylized version of the diagnostic and treatment process for a known disease is shown in Fig. 1. A patient's visit to a clinic is met with an assessment that includes visual acuity tests and collecting demographic information. This provides Best Corrected Visual Acuity (BCVA) scores, Patient ID, and Eye ID among other data. We term these as *clinical labels*. Next, the patient undergoes diagnostic imaging that includes Fundus and OCT scans. Finally, a trained practitioner interprets the diagnostic scans for known *biomarkers* for diseases. The authors in (1) describe biomarkers as objective indicators of medically quantifiable characteristics of biological processes which are often diseases. The biomarkers along with the scans and clinical labels are used to assess the presence and severity of a patient's disease and a recommendation of a treatment is provided. If the recommendation is yes, the patient is treated and asked to visit again after a gap.

---

*Equal Contribution

36th Conference on Neural Information Processing Systems (NeurIPS 2022) Track on Datasets and Benchmarks.

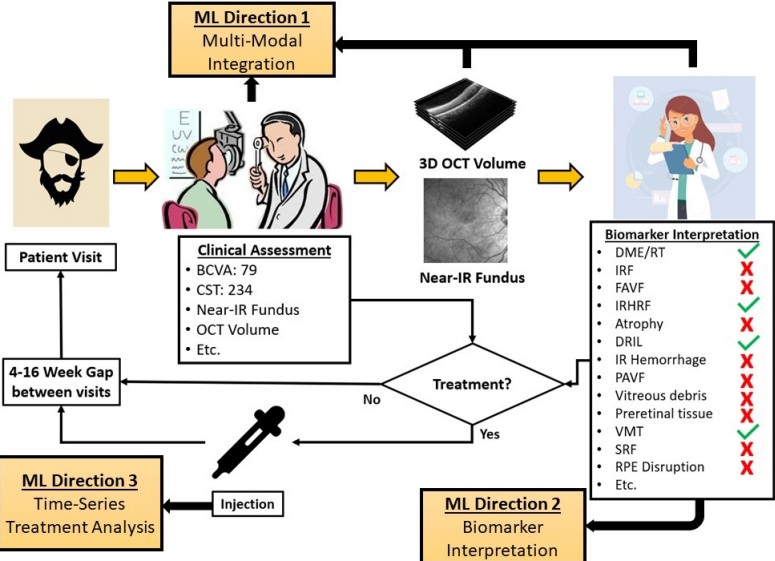

Figure 1: Complete summary of data collection process for the `OLIVES` dataset and potential research directions for the machine learning community.

A number of Machine Learning (ML) techniques have sought to either automate or interpret individual processes within Fig. 1. We annotate three such ML research directions within the pipeline for clinically aiding and monitoring disease diagnosis and treatment. The first direction involves assessing multi-modal data for clinical applications including predicting disease states. The second direction is an interpretation of biomarkers. Biomarkers act as intermediary data between medical scans and disease diagnosis that aid clinical reasoning. The last direction is analyzing time-series treatment data across the treatment period. This direction aids initial treatment prescription and patient monitoring. To the best of our knowledge, no existing dataset provides access to data that promotes all three stated research directions for the clinical process from Fig. 1. In this paper, we introduce the Ophthalmic Labels for Investigating Visual Eye Semantics (`OLIVES`) dataset that provides structured and curated data to promote holistic clinical research in ML for ophthalmic diagnosis.

**Clinical studies for `OLIVES` dataset**    The `OLIVES` dataset is derived from the PRIME (2) and TREX DME (3; 4; 5; 6) clinical studies. Both the studies are prospective randomized clinical trials that were run between December 2013 and April 2021 at the Retina Consultants of Texas (Houston, TX, USA). Prospective trials refer to studies that evaluate the outcome of a particular disease during treatment. PRIME evaluates Diabetic Retinopathy (DR) and TREX-DME evaluates Diabetic Macular Edema (DME). The trials provide access to near-IR fundus images and OCT scans along with de-identified Electronic Medical Records (EMR) data of 96 patients across an average of 66 weeks. Biomarkers are retrospectively added to this data by experienced graders upon open adjudication.

**Challenging dataset for ML research**    While challenges in natural images are generally contrived by intervening on top of data (7; 8; 9), the complexities in ophthalmic datasets arise because of issues in data collection, inversion, representation and annotation. (10). `OLIVES` data modalities range from 1-dimensional numerical values (BCVA, Patient ID), vectorized biomarkers, 2-dimensional fundus images, and 3-dimensional scans (optical coherence tomography). Moreover, some of this data is objectively measured through instruments from patients (fundus, OCT), subjectively collected through eye tests (BCVA), while other data is interpreted and openly adjudicated through images (biomarkers). The variation within scans between visits can be minimal while the difference in manifestation of the same disease between patients may be substantial. This is shown in Fig. 2. The domain difference between OCT scans can arise due to pathology manifestation between patients (Fig. 2a and Fig. 2b), clinical labels (Fig. 2c), and the visit along the treatment process when the scan is taken (Fig. 2d). `OLIVES` provides access to these challenging data modalities that allow for innovative ML algorithms.

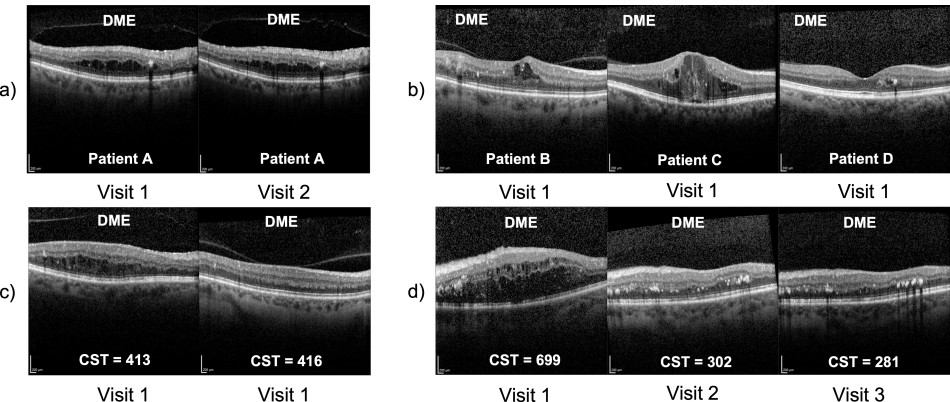

Figure 2: An illustration of some challenges within the dataset. In a) variation in OCT belonging to the same patient at different visits is minimal. In b) varied disease manifestations are among different patients. c) At times CST values are very similar but the OCT is visually dissimilar. d) Shows that CST values gradually decrease as time progresses.

**Contributions and significance of the dataset**  The OLIVES dataset is curated to foster research in ophthalmic ML. The retrospective additions to the OLIVES dataset from its base clinical trials and its ensuing contributions include:

1. Sixteen biomarker labels are added to the OCT scans of every first and last visit of all patients. We experimentally validate the necessity of biomarkers and provide benchmarks in Sections 4.1 and 4.3. Along with biomarkers, OLIVES provides access to fundus, OCT scans, clinical labels and DR/DME diagnosis, thereby creating an ideal benchmarking mechanism for ophthalmic ML.

2. We curate the clinical labels that have known correlations between the four data modalities. These include Best Corrected Visual Acuity (BCVA), Central Subfield Thickness (CST), Patient ID, and Eye ID. We demonstrate its utility for medically-grounded contrastive learning where augmentations are based on contrasting between clinical labels in Section 4.2. Hence, OLIVES dataset promotes research in core and emerging ML paradigms.

3. The data and labels are made accessible to non-medical professionals. Biomarkers act as expert-annotated and interpretable visual indicators of diseases within OCT scans. The original labels from the clinical trials along with their data sheets are provided in Appendix B.5.2. Additionally, an ML-specific set of labels which is relevant to the three mentioned research directions in Fig. 1, is provided in Appendix D.3.

## 2   Related Works

**Ophthalmology datasets**  A number of publicly available ophthalmology datasets individually tackle each of the clinical modalities that exist in the OLIVES dataset. The authors in (11) provide a survey of 94 existing open access ophthalmic datasets. Among 54 of the 94 datasets, the underlying data is that of fundus images. 19 of the remaining datasets contain 3-dimensional OCT scans. The OCT scans provide structural information that enhances the performance of machine learning algorithms (11). Only three of the 94 considered open access datasets provide both OCT and fundus image modalities. The authors in (1) provide 650 OCT slices from a single volume. These are insufficient to leverage the data intensive machine learning algorithms to provide generalizable results. In contrast, the OLIVES dataset has 78, 189 slices taken from 1268 volumes. (12) provide OCT and fundus data from 50 healthy patients. However, these are all for healthy eyes and disease manifestation is not observed. Other datasets including (13) contains OCT scans for four OCT disease states: Healthy, Drusen, DME, and choroidal neovascularization (CNV). (14) and (15) introduced OCT datasets for age-related macular degeneration (AMD). (16) contains OCT scans labeled with segmentation of regions with DME. However, these datasets do not possess comprehensive clinical information or a wide range of expert-annotated biomarkers. A complete overview that considers clinical labels, biomarkers, disease labeling, and time-series analysis is provided in Tables 5 and 6.

We refer to (11) to compare other statistics including number of image scans and applicability of existing datasets against `OLIVES`.

**Machine learning techniques on ophthalmic data**    A number of works have separately addressed the research directions identified in Fig. 1. The authors in (17) proposed transfer learning to screen for relative afferent pupillary defect due to lack of comprehensive data. (18) showed that transfer learning methods could be utilized to classify OCT scans based on the presence of key biomarkers. (19) introduced a dual-autoencoder framework with physician attributes to improve classification performance for OCT biomarkers. (20) expanded previous work towards segmentation of a multitude of different biomarkers and referred for different treatment decisions. Other work has demonstrated the ability to detect clinical information from OCT scans which is significant for suggesting correlations between different domains. (21) showed that a model trained entirely on OCT scans could predict the associated BCVA value. Similarly (22) showed that values such as retinal thickness could be learned from retinal fundus photos. The `OLIVES` dataset provides a standardized benchmark to conduct research across applications, data modalities and machine learning paradigms.

Table 1: High-level overview of the `OLIVES` Dataset. The modality column details the type of data. The columns "Per Visit" and "Per Eye" indicate the amount of data in each modality on a respective visit or eye. $N_P$ is the number of visits that a patient $P$ takes to the clinic. The statistics across all eyes across all visits are shown in the Total Statistics column. Biomarkers are binary values, clinical labels are integers, fundus are 2D images, and OCT are 3D slices.

| OLIVES Dataset Summary | | | | |
|---|---|---|---|---|
| Modality | Per Visit | Per Eye | Total Statistics | Overview |
| | | | | **General:** |
| OCT | 49 | $N_P$*49 | 78189 | 96 Eyes, Visits every 4-16 weeks, |
| | | | | Average 16 visits and 7 injections/patient |
| Fundus | 1 | $N_P$ | 1268 | **Clinical Labels obtained every visit:** |
| | | | | BCVA, CST, Patient ID, Eye ID |
| Clinical | 4 | $N_P$*4 | 5072 | **Biomarkers labeled:** |
| | | | | IRHRF, FAVF, IRF, DRT/ME PAVF, VD, |
| Biomarker | 16 | 1568 | 150528 | Preretinal Tissue, EZ Disruption, IR Hemmorhages, |
| | | | | SRF, VMT, Atrophy, SHRM, RPE Disruption, |
| | | | | Serous PED |

## 3    OLIVES Dataset

Statistics regarding the quantity of images and labels can be found in Table 1. The `OLIVES` dataset is derived from the PRIME and TREX-DME trials. At every visit for each patient, ocular disease state data (DR/DME), clinical labels including BCVA, CST, Patient and Eye ID, and detailed ocular imaging including OCT, and fundus photography were obtained per the protocol in Section B.5.2. This procurement of data continues across $N_P$ visits for every patient, where $N_P$ is the number of visits by a patient $P$. For instance, 3D longitudinal scans of the eye provide 49 OCT scans per patient per visit. Across $N_P$ visits where $P$ can be any one of 96 patients, the total number of OCT scans in the dataset is 78, 189. Note that on every visit, each patient undergoes testing to determine the requirement of a treatment per the clinical protocol described in Appendix D.1. Biomarkers are retrospectively added to each slice in the OCT scans for the first and last visits. Table 1 also indicates the total number of eyes, average number of visits and injections, and the time between visits.

### 3.1    Biomarker Generation

After the clinical data collection process, we retrospectively provide additional insight into the OCT scans by providing corresponding biomarker labels. Biomarkers are quantifiable characteristics of biological processes in the eye. In this paper, the biological processes are diseases and biomarkers indicate the presence or absence of such diseases. Under limited circumstances, the authors in (1) suggest that biomarkers can be surrogate endpoints in clinical trials. However, they caution against doing so unless the underlying clinical trial is specifically meant for the study. In both the PRIME and TREX DME studies, biomarkers are retrospectively labeled. As such, biomarkers may indicate the presence of diseases, but are not causal to these diseases. Hence, biomarkers are different from visual causal features from (23) or causal question-based analysis in (24) or causal factor analysis in (25).

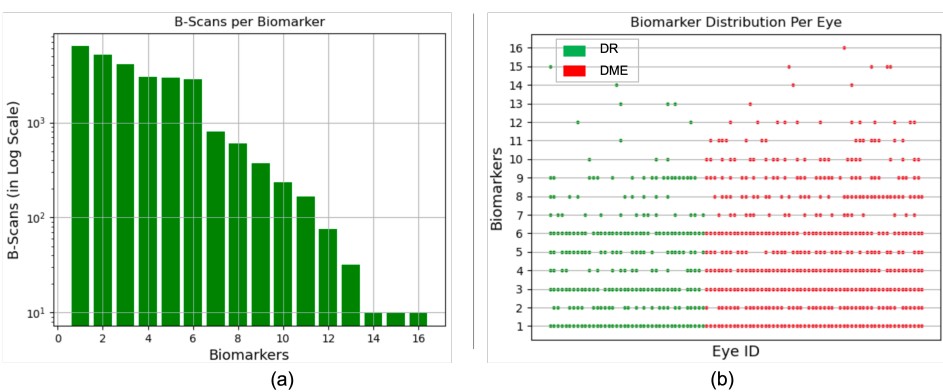

Figure 3: (a) Histogram of the number of scans per biomarker. (b) Unique biomarkers per Eye ID.

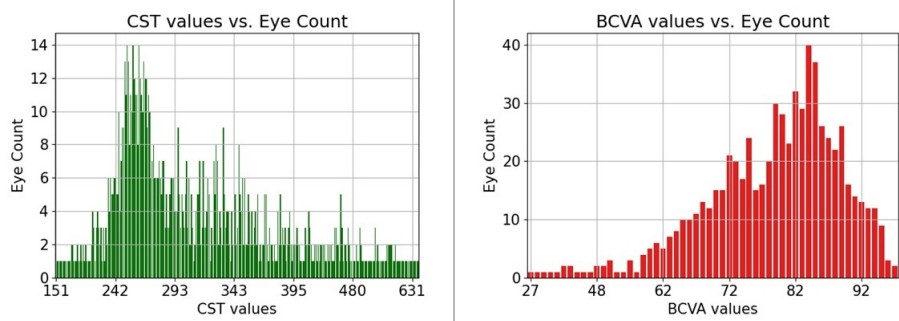

Figure 4: Distribution of CST and BCVA labels in `OLIVES` dataset based on number of eyes associated with each clinical value.

In the PRIME and TREX DME studies, images, clinical information, and biomarker labels were retrospectively collected at the Retina Consultants of Texas (Houston, TX, USA). This study was approved by the Institutional Review Board (IRB)/Ethics Committee and adheres to the tenets of the Declaration of Helsinki and Health Insurance Portability and Accountability Act (HIPAA). Informed consent was not required due to the retrospective nature of the study. A trained grader performed interpretation on OCT scans for the presence of 16 different biomarkers including: intraretinal hyperreflective foci (IRHRF), partially attached vitreous face (PAVF), fully attached vitreous face (FAVF), intraretinal fluid (IRF), and diffuse retinal thickening or macular edema (DRT/ME). A full list of the biomarkers as well as their characteristics is provided in Section B.5.1. The full form of the abbreviations are given in Table 7. These biomarkers are chosen because of their visual attributes that correlate with presence or absence of disease states. The trained grader was blinded to clinical information whilst grading each of 49 horizontal OCT B-scans of both the first and last study visit for each individual eye. Open adjudication was done with an experienced retina specialist for difficult cases. In total, there are $9408$ OCT scans that consist of a $16 \times 1$ biomarker vector where $1$ indicates the presence of the corresponding biomarker and a $0$ indicates its absence. We provide a histogram of the number of scans (`y-axis`) against their respective biomarkers in Fig. 3a. Note that the `y-axis` is in log-scale. We also depict the eye ID against the biomarkers in Fig. 3b. The green dots are eyes that indicate the presence of the corresponding biomarker on the `y-axis` that are diagnosed with DR. The red dots are for DME. It can be seen that a number of eyes have overlapping biomarkers even between diseases. Hence, biomarkers in isolation are insufficient to diagnose disease states, strengthening the case for multi-modal data.

## 3.2 Clinical Labels

Within the `OLIVES` dataset, we have explicit clinical information regarding the Best Central Visual Acuity (BCVA), Central Subfield Thickness (CST), and identity of the eye. ETDRS best-corrected visual acuity (BCVA) is a visual function assessment performed by certified examiners where a standard vision chart is placed 4-meters away from the patient. The patient is instructed to read

the chart from left to right from top to bottom until the subject completes 6 rows of letters or the subject is unable to read any more letters. The examiner marks how many letters were correctly identified by the patient. Central subfield thickness (CST) is the average macular thickness in the central 1-mm radius of the ETDRS grid. Both BCVA and CST are coarse measurements over the eye as opposed to Biomarkers that exist for fine-grained longitudinal slices of the eye. BCVA can range from $0 - 100$ and CST from $100 - 1300$. We show in Fig. 4 the number of eyes (y-axis) that have the associated value (x-axis) for both BCVA and CST. This graph shows that our dataset has a wide variation in terms of range of clinical values across a multitude of eyes in the dataset. This is advantageous as it shows the dataset is not biased to any specific range of values or localized to single eye instances. A full list of all clinical labels present in PRIME and TREX-DME clinical trials are shown in Section B.5.2. No personally identifiable information was included in compliance with HIPAA regulations.

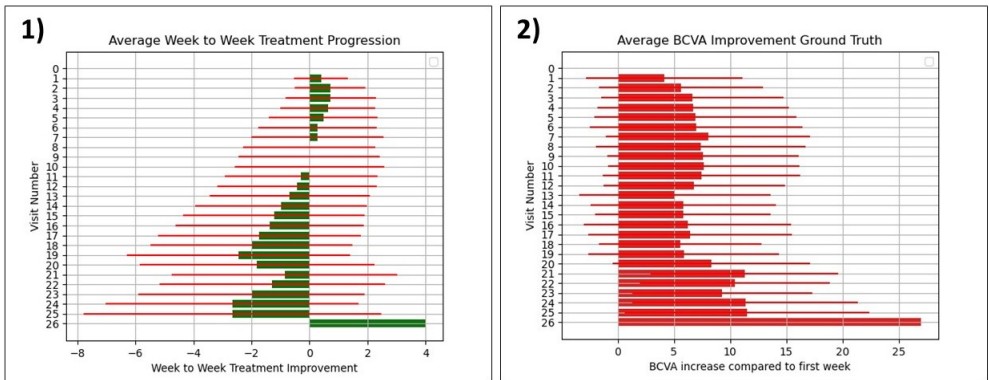

Figure 5: 1) A plot of average number of visits by patients that were an improvement or deterioration from previous week. Red bars indicate the standard deviation across all patients. 2) Plot of average change in BCVA with respect to the first week.

### 3.3   Time-series data

A core novelty of the dataset is that data exists for each patient visit across a defined period of time. As a result, it is possible to analyze trends in the collected imaging and clinical data over the visiting period of the patient. This is shown in Fig. 5 with an overall progression analysis shown by the bar graphs. Graph 1 indicates whether on average there was an improvement from the previous visit. This was computed by assigning a value of 1 for improvements and -1 for deterioration. This is accumulated every visit and the average across all patients is calculated on a per visit basis. From this plot, it can be observed that in the dataset, eyes generally improve on every visit until about the tenth visit. However, graph 2 n Fig. 5 shows that while visit to visit improvement declines, the overall improvement when compared with the first visit is generally substantial. This graph was computed by taking the difference between the current visit's BCVA and first visit's BCVA and averaging across all patients. The statistics of the number of patients every visit and visualizations of patient treatment is shown in Figs. 8 and 9 respectively.

### 3.4   Interaction between Data Modalities

Clinical labels correspond to measurements that pertain to the entire visual system, including the visual mechanism in the eye. These measurements give an overview of the health of the eye, but they do not enable fine-grained analysis of structures that exist within the eye. Biomarker labels exist at the longitudinal slice level. They are detailed labels for every slice of the eye and provide a fine-grained analysis of the biological structures that exist within the eye. Clinical studies such as (26) and (27) suggest that measured clinical labels can act as indicators of structural changes that manifest themselves in OCT scans and fundus images as well as the severity of disease associated with the patient. For example, visually, it can be observed that OCT scans with the same BCVA values exhibit more common structural characteristics than scans with different BCVA values. Furthermore, all data modalities exhibit visual, structural and clinical changes across the treatment period. OLIVES dataset

allows for exploiting these correlations between OCT, fundus, clinical labels, biomarkers, diseases, and treatment states.

## 4 Clinical Applications

With the multitude of modalities that exist within the `OLIVES` dataset, there is potential for research in a wide variety of ML applications. Within this section, we focus on applications, and benchmarks, that showcase key features of the dataset identified from Fig. 1, but acknowledge that other novel setups and formulations of the problem are possible and intended. These applications include multi-modal integration of OCT scans and biomarker/clinical labels, biomarker detection and interpretation using contrastive learning, and time-series treatment analysis.

### 4.1 Multi-Modal Integration Between OCT and Biomarkers/Clinical Labels

Table 2: Benchmark results for DR/DME detection.

| Experiments | Model | Balanced Accuracy | Specificity | Sensitivity |
|---|---|---|---|---|
| OCT | R-18 | 70.15% $\pm$ 4.69 | 0.608 | 0.794 |
| Clinical | MLP | 75.49% $\pm$ 1.98 | 0.758 | 0.751 |
| Biomarker | MLP | 79.87% $\pm$ 3.03 | 0.826 | 0.771 |
| OCT + Clinical | R-18 + MLP | 75.92% $\pm$ 3.05 | 0.566 | 0.952 |
| OCT + Biomarker | R-18 + MLP | 82.33% $\pm$ 3.59 | 0.742 | 0.904 |

**Baseline Detection of DR/DME with OCT**    Since biomarkers are only available for the first and last clinical visits, we use the corresponding OCT at those visits for this baseline analysis. The entire dataset is partitioned by eyes into train, test and validation splits. Additional details about train/test/validation splits is in Appendix C.1. We evaluate performance with balanced accuracy, precision and recall performance metrics. The results for the baseline OCT model is shown in the first row of Table 2. Additional results showing specificity and sensitivity are in Table 9 in the Appendix. This and subsequent experiments are conducted using multiple random seeds for DR/DME detection and an average score and standard deviation is reported for balanced accuracy.

**Supervised Learning with Clinical Labels**    We aim to use clinical labels as an additional modality to aid the baseline model. However, to determine the suitability of this auxiliary data type, we first evaluate its impact on the classification of DR and DME. To do this we first find all unique clinical labels present in the dataset with their associated disease labels. Then, we create a training set with 70% of these clinical labels along with test and validation sets of 20% and 10% proportions respectively. This yields 1107 unique clinical labels for training, 306 for testing and 122 for validation. Within the test set, half the samples are DR and the remaining DME. The second row on Table 2 shows that CST and BCVA used as clinical features are more effective than the unimodal OCT baseline for DR/DME detection.

**Supervised Learning with Biomarkers**    We perform a similar analysis as described in supervised learning with clinical labels but using biomarkers as features. Hence, we substitute the clinical labels with biomarkers to characterize the diseases. There are 286 unique biomarker label features among which 200, 58, 28 samples are used for train, test and validation sets respectively. From the third row in Table 2, we observe that using biomarkers on their own leads to a 9.72% increase in DR and DME classification over baseline results.

**Multi-Modal Learning with OCT and Clinical Labels**    Having seen that clinical labels are more effective than the baseline model at DR/DME classification, we now investigate how to use the clinical label modality to aid the OCT model. Clinical labels and OCT are independently given as input to their models as described previously. We optimize both models jointly with a loss function that allows knowledge, in the form of logits, from the clinical model to guide the optimization of the OCT model. A detailed description of this optimization scheme can be seen in Appendix C.1. During testing, only the OCT model, having been optimized jointly with the other model, is used to classify the disease states. The fourth row of Table 2 shows that clinical labels also aid the OCT model at characterizing the diseases albeit not the most effectively.

Table 3: Benchmark of the performance of supervised contrastive training on images with clinical and biomarker data. The standard deviations are shown in Table 10.

| Method | Biomarkers | | | | | | | | | | Metrics | | |
|---|---|---|---|---|---|---|---|---|---|---|---|---|---|
| | IRF | | DRT/ME | | IRHRF | | FAVF | | PAVF | | AUROC | Average Specificity | Average Sensitivity |
| | Accuracy | F1-Score | Accuracy | F1-Score | Accuracy | F1-Score | Accuracy | F1-Score | Accuracy | F1-Score | | | |
| PCL [28] | **76.50%** | 0.717 | 80.11% | 0.761 | 59.10% | 0.683 | 76.30% | 0.773 | 51.40% | 0.165 | 0.767 | 0.741 | 0.604 |
| SimCLR [29] | 75.13% | 0.716 | 80.61% | 0.772 | 59.03% | 0.675 | 75.43% | 0.761 | 52.69% | 0.249 | 0.754 | 0.747 | 0.614 |
| Moco V2 [30] | 76.00% | **0.720** | 82.24% | 0.793 | 59.60% | 0.692 | 75.00% | 0.784 | 52.69% | 0.211 | 0.770 | 0.762 | 0.651 |
| Eye ID | 72.63% | 0.674 | 80.20% | 0.778 | 58.00% | 0.674 | 74.93% | 0.725 | **65.56%** | **0.588** | 0.767 | **0.776** | 0.656 |
| CST | 75.53% | 0.720 | **83.06%** | **0.811** | **64.30%** | **0.703** | 76.13% | 0.766 | 62.16% | 0.509 | **0.790** | 0.772 | **0.675** |
| BCVA | 74.03% | 0.701 | 80.27% | 0.770 | 58.8% | 0.672 | **77.63%** | **0.785** | 58.06% | 0.418 | 0.776 | 0.713 | 0.645 |

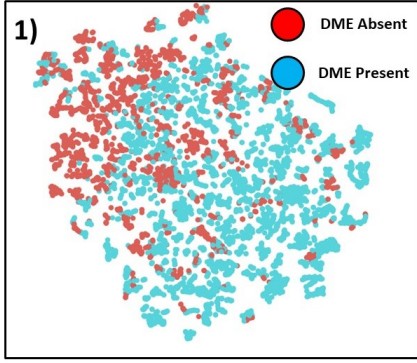 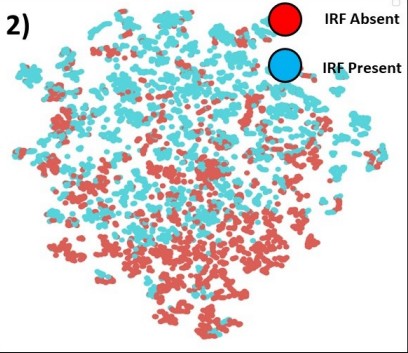

Figure 6: T-SNE visualization of the OLIVES Biomarker test set labeled by the presence or absence of DME and IRF. We can effectively achieve an embedding space that is separable with respect to biomarkers 1) DME and 2) IRF.

**Multi-Modal Learning with OCT and Biomarkers**  In like manner, we investigate the impact that biomarkers as features can have on aiding an OCT model to classify the two diseases. Each model is fed their input modality and optimized jointly using the same loss detailed in Appendix C.1. During testing, only the OCT model is used for inference. The final row in Table 2 shows that this is the most effective technique that significantly improves all baseline classification metrics.

## 4.2 Biomarker Interpretation with Contrastive Learning

Due to the prohibitive costs of expert-annotated biomarker labels, contrastive learning (31; 29; 32) approaches have garnered attention because of their state of the art self-supervised performance. These approaches generally create a representation space through minimizing the distance between positive pairs of images and maximizing the distance between negative pairs. Traditional approaches, like SimCLR (29), generate positives from augmentations of a single image and treat all other images in a batch as negatives. More modern approaches like Moco v2 (30) incorporate a queue system for additional negative samples while extensions of this include PCL (28) that introduce a clustering approach on the representation space. While these approaches have shown promising results on natural images, such augmentations are unrealistic for medical images that rely on fine-grained changes within OCT scans to detect diseases. Instead, we propose using clinically relevant labels as a means to better choose positive pairs. Since OLIVES provides a larger pool of clinical labels than biomarker labels, this task fits well within the scope of the dataset. Hence, OLIVES enables research into novel and multi-modal contrastive learning strategies. We implement one such strategy through reformulating the supervised contrastive loss (33) in a clinical context as discussed in (34) with related work located at (35; 36). Implementation details are provided in Section C.2.

We train a Resnet-18 (37) encoder with the clinically labeled data using the clinically aware supervised contrastive loss. After training the encoder with supervised contrastive loss, we freeze the weights of the encoder and append a linear layer to its output. This linear layer is trained using cross-entropy loss to distinguish between the presence or absence of the biomarker of interest in the OCT scan. Not all of these biomarkers exist in sufficiently balanced quantities to train a model to identify their presence or absence within an image. Hence, we use five biomarkers that fit this criteria in this study. The training details are presented in Section C.2. We compare this method with three state of the art self-supervised algorithms in Table 3. We evaluate performance in terms of individual biomarker accuracy and f1-score as well as in the setting where the goal is to simultaneously perform a multi-

label classification of biomarkers. Performance is measured by average AUROC, specificity, and sensitivity. We observe that a training strategy that chooses positives based on the clinical data Eye ID, BCVA, and CST outperforms baseline self-supervised methods in both a multi-label classification task as well as individual biomarker detection performance. While the results in Section 4.1 make use of correlations between the biomarkers and clinical labels with disease states, these results depict the correlations between the label modalities.

In Figure 6, we visualize the test set t-SNE embeddings of two different biomarkers from a model trained using the BCVA clinical label. We observe that even without any fine-tuning on the actual biomarker label of interest, we are able to get an embedding space where the absence and presence of DME and IRF form distinct clusters. This gives credence to the idea that there exists relationships between the biomarker and clinical label domains as training on only clinical labels leads to a separable space within the biomarker domain.

### 4.3 Time-Series Treatment Analysis

The multi-modal nature of OLIVES dataset allows for a large combination of experimental setups to analyze treatments. We present two experimental manifestations based off the temporal nature of the data: a) Predicting visit-by-visit successive treatment effects and b) Predicting the final ocular state using Biomarkers. A key metric used to evaluate treatment progression or regression is BCVA. At each visit to the clinic, patients' ocular disease states are evaluated and BCVA and other clinical labels are recorded. From a machine learning perspective, this motivates an analysis of treatment effect over consecutive weeks to predict how BCVA scores will change based on the state of the eye captured via OCT or Fundus. We detail the exact experimental procedure in Appendix C.3. We evaluate the performance of this strategy on both fundus images and 3D OCT volumes. We use a Resnet-18 (37), ResNet-50 (37), DenseNet-121 (38), EfficientNet (39), and Vision Transformer (40) (using a patch size of 32, 16 transformer blocks, 16 heads in multi-attention layer). For the OCT volumes, we utilize a version of each architecture that uses three-dimensional convolution layers. Performance in both modalities is reported in Table 4. We observe that the model is able to learn distinguishing features between the two classes, with better performance when using the OCT volumetric data. Additionally, we present results for predicting the final state of $16 \times 1$ biomarker vector given the initial biomarker vector for individual patients in Fig. 10. Similar to the week-wise case, these results indicate correlation among multiple modalities as well as the ability of ML algorithms to predict ocular states given treatment.

Table 4: Benchmark Performance of predicting treatment effects from time-series Fundus and OCT data.

| Model | Image Modality | Accuracy | Precision | Recall |
|---|---|---|---|---|
| ResNet-18 | Fundus | 55.19% $\pm$ 10.9 | 0.256 | 0.343 |
| | OCT Volume | 57.59% $\pm$ 9.51 | 0.359 | 0.326 |
| ResNet-50 | Fundus | 48.73% $\pm$ 13.3 | 0.372 | 0.3296 |
| | OCT Volume | 57.70% $\pm$ 9.1 | 0.301 | 0.1826 |
| DenseNet-121 | Fundus | 53.00% $\pm$ 8.9 | 0.273 | 0.259 |
| | OCT Volume | 54.75% $\pm$ 4.92 | 0.219 | 0.188 |
| EfficientNet | Fundus | 56.06% $\pm$ 4.85 | 0.292 | 0.217 |
| | OCT Volume | 60.65% $\pm$ 4.09 | 0.3613 | 0.1633 |
| ViT | Fundus | 55.01% $\pm$ 3.27 | 0.285 | 0.350 |

## 5 Discussion and Conclusion

**Domain Difference and Adaptation in Multi-Modal Data** The data in OLIVES is derived from two studies. As mentioned in Section 1, the domain difference in ophthalmic data can arise from sources such as treatment, disease manifestation, and clinical labels. In natural images, one source of domain difference is the equipment used for imaging. In PRIME and TREX studies, the same imaging and grading modalities, the Heidelberg Spectralis HRA+OCT software, is used in the same clinic. We provide extensive experiments in Appendix C.5 and C.6 to characterize possible domain differences on OLIVES. In Table 11, we show that the biomarker detection results when trained and tested on PRIME trial is lower than when trained with TREX and tested on PRIME. This is because a longer treatment period on TREX dataset provides more diverse data that is conducive for

training ML algorithms. Intuitively, this suggests that treatment causes domain shift in data, which is illustrated in Table 12. Training and testing within the first week data provides the best results for biomarker detection. This analysis is further expanded in Fig. 11. Rather than showing domain difference, we adapt between the first and last visit domains. Specifically, we use a part of the last visit data to train with the first visit data and show that: a) adapting between OCT scans before and after treatment is possible, and b) the addition of biomarkers increases the results for diagnosis of DR/DME. Hence, OLIVES provides data modalities that promotes research in treatment-based domain difference and adaptation in medical data.

**Dataset Limitations, Societal Impact, and Ethical Concerns**   The OLIVES dataset is derived from two clinical studies conducted from only one U.S. clinic. While there is a range in the age, ethnicity and racial demographics within the cohorts, this range is only limited to one geographical location. Hence, an end-to-end system can be biased. To mitigate this limitation, we provide links to existing open access ophthalmic datasets in Appendix B.4 that are collected from other parts of the world. While none of these datasets are as rich as our own in terms of numbers, modalities, or labels, they can be used to modularly test algorithms. We present one such result in Table 13 and show that combining datasets allows for higher results. The PRIME and TREX trials are randomized clinical studies with the goal of comparing different treatment regimens. These studies aim to find the best practices for how and when they should treat patients to get the most optimal outcomes. However, there are no control groups within the studies that did not receive treatment. While this is common in clinical trials (6), it adds a new challenge to ML-focused research of time-series analysis. We list datasets that provide healthy images in Appendix B.4 to complement OLIVES. We believe that a combination of datasets taken over multiple geographical regions, times, and disease states is essential to construct generalizable and ethical ML models. ML models can potentially amplify existing inequalities within healthcare access (41). For instance, the data in OLIVES is collected from December 2013 to April 2021, which implies the participants had the time and means to be part of these trials. This may not always be the case for disadvantaged groups. Hence, any benefit that machine learning could provide will be restricted to small subsets of society unless thought is put into preventing this disparity. Hence, a careful analysis of potential concerns is required to use OLIVES and any other dataset to enrich the functionality and adaptability of machine learning algorithms in everyday lives.

**Conclusion**   We introduce the OLIVES dataset to bridge the gap between existing ophthalmic datasets and the clinical diagnosis and treatment process. OLIVES provides curated and contained data that can be used for clinical interpretation of biomarkers, clinical reasoning regarding disease prediction, multi-modal integration of ophthalmic data and treatment monitoring through time-series analysis. Also, we propose and benchmark medically-grounded contrastive learning strategies that are possible because of the presence of correlated multi-modal data within the introduced dataset. The OLIVES dataset opens new frontiers for training holistic and medically-relevant ML frameworks that mimic the clinical diagnosis pipeline for ophthalmic studies.

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
