# OpenReview forum: "OLIVES Dataset: Ophthalmic Labels for Investigating Visual Eye Semantics"
_NeurIPS.cc/2022/Track/Datasets_and_Benchmarks — NeurIPS 2022 Datasets and Benchmarks _

### Official Review · Reviewer_fPn3 · 2022-07-26
**provided Ophthalmic dataset with OCT and near-IR fundus images that includes clinical labels, biomarker labels, disease labels, and time-series patient treatment information from associated clinical trials**

**Rating:** 5
**Confidence:** 5
**Correctness:** yes
**Clarity:** yes

**Strengths:**

The Paper is well written and correctly addresses the problem statement.
The Paper has introduced a dataset with three modalities and shown its scope in the field of ML.


**Weaknesses:**

1. The size of the dataset introduced is small.
2. The data is collected from two trials, PRIME and TREX. The authors have not mentioned the differences, which may/may not affect the model evaluation with collected samples.
3. The significance of the clinical features such as BCVA etc., should have been better explained to draw the comparison across the modalities.
4. Clinical labels and biomarkers are associated with each eye. How is the relation across the two modalities developed for the datasets? Per my understanding, there should be a 'Patient id' with 'left' and 'right' eyes and corresponding clinical labels and biomarkers associated with each sample (eye).
5. Are the mentioned three modalities correspond to the same patient? This means there are three samples across three modalities for each patient
6. Results in terms of sensitivity and specificity are missing, which are important for evaluating the ML model for disease diagnosis.
7. In Table 3, many inputs used to train the model have shown a random accuracy for binary classification. This proves the insignificance of these features and contradicts the author's claims. Similar results are found in table 7 in the supplementary.

8. Data collected from a single centre might encourage data bias.


**Additional Feedback:**



**Documentation:**

yes

**Ethics:**

Consent from patients is taken for the clinical trials.

**Relation To Prior Work:**

yes

**Summary And Contributions:**

The authors provided an Ophthalmic dataset with OCT and near-IR fundus images, including clinical labels, biomarker labels, disease labels, and time-series patient treatment information from associated clinical trials. The authors introduced the OLIVES dataset to bridge the gap between existing ophthalmic datasets and the clinical diagnosis and treatment process.

---

> ### Author Response · Authors · 2022-08-24
> **Addressing size of the dataset, Domain shift and Sensitivity/Specificity Experiments**
>
> We thank the reviewer for their comments and address each of them below.
>
> **The size of the dataset introduced is small.**
>
> In Appendix B.1 and Tables 4 and 5, we compare the OLIVES dataset to other related ophthalmic datasets that includes the number of images and eyes. Within the context of comparable ophthalmic datasets, the OLIVES dataset is the second largest in terms of total number of images, while having a larger array of relevant tasks. Additionally, the OLIVES dataset contains time-series data that is not explored in any other large-scale dataset. This data is curated from clinical trials that extended from December 2013 - April 2021. Please note that this is not uncommon for datasets collected with time-series labels. The dataset in [1] is also the outcome of 7 years worth of trials. Hence, these datasets require large time and personnel commitments to obtain and curate. We firmly believe that a combination of datasets taken over multiple geographical regions, times, and disease states are essential to construct generalizable ML models. To this end, we incorporated the Kermany dataset [13 in paper] in our experiments in Appendix C.7, Table 13. The diversity within both the datasets allowed for higher results as compared to Natural image developed contrastive learning techniques like SimCLR, PCL, and MoCo V2. However, it is still lesser as compared to using the clinical labels from OLIVES thereby validating the need for multi-modal data.
>
> **The data is collected from two trials, PRIME and TREX. The authors have not mentioned the differences, which may/may not affect the model evaluation with collected samples.**
>
> We thank the reviewer for these comments. We performed and included a number of experiments to address this question. We answer this question qualitatively and quantitatively:
>
> 1. Qualitative: A full discussion of each of the two trials are presented in appendix D.1. PRIME studies DR while TREX studies DME diseases. The eyes in TREX are studied over three years and have more severe conditions than those present in PRIME. For this reason, combining the two data allows for a more complete distribution in terms of the structural specifications of the eye. This is opposed to having a single clinical trial where most of the images are healthy or most of the images are diseased. Moreover, combining the studies allows for studying differentiating factors between DR/DME instead of between healthy vs single-diseased. Logistically, both trials are conducted at the exact same clinic within a similar population group using the same imaging setup. Hence, any domain difference that exists between trials is due to the disease manifestation, treatment over the course of three years, and multi-modal data. We expand on this in Fig. 2 and Line 60 in the Introduction.
>
> 2. Quantitative: In Table 11, we perform additional experiments where we perform intra-trial vs inter-trial experiments. Intra-trial refers to within PRIME and within TREX experiments - train and test within respective trials. Inter-trial refers to training and testing on different trials. Since they study separate diseases, we use the biomarker detection (which is present between diseases) using contrastive learning as the experimental setup for this study. As can be expected the best results are obtained when training and testing on TREX - this is because of diversity in TREX data due to the larger clinical trial window of 3 years. Interestingly, the inter-trial results when training on TREX and testing on PRIME is higher than intra-trial training and testing on PRIME.
>
> However, domain shift can also occur due to treatments. Both time and treatment changes the manifestation of diseases, thereby domains. We perform two additional experiments to analyze and account for this shift:
>
> 1. We experiment by training and testing before and after treatments (first and last visits) and show that ML models are susceptible to this domain shift. This is discussed in Appendix C.5 and Table 12.
>
> 2. We perform domain adaptation experiments where we take a part of the Final week data and append it to the first week data (10%, 20%, and 30%) and perform DR/DME detection. The balanced accuracy with only OCT images shows an increase of 35% in Fig. 11. However, adding biomarkers to this setup adds an additional average of 10% to the results, showcasing the value of multi-modal data.
>
> We discuss this in the Discussion and Conclusion Section - Lines 303 onwards.
>
> **Results in terms of sensitivity and specificity are missing, which are important for evaluating the ML model for disease diagnosis.**
>
> In the revised version of the paper, we have sensitivity/specificity in Tables 2 and 3, and precision/recall in the Appendix Tables 8 and 10.
>
> [1] Rivail, Antoine, et al. "Modeling disease progression in retinal OCTs with longitudinal self-supervised learning." International Workshop on PRedictive Intelligence In MEdicine. Springer, Cham, 2019.

---

> > ### Author Response · Authors · 2022-08-24
> > **Significance of clinical labels; Relationship between labels and organization within the label files; Addressing dataset bias**
> >
> > We continue the rebuttal from the previous comment.
> >
> > **The significance of the clinical features such as BCVA etc., should have been better explained to draw the comparison across the modalities.**
> >
> > We thank the reviewer for this comment and have addressed it across the revised paper. Specifically,
> >
> > 1. We define biomarkers and their relationship with diseases before describing their labeling in Section 3.1, Line 129.
> >
> > 2. We expand on the significance of the clinical labels, their meaning and ranges in Section 3.2. Specifically, clinical labels provide coarse measurement for the entire eye while biomarkers provide fine-grained labels on 3D longitudinal slices of the eye.
> >
> > 3. We add a new Subsection 3.4 that details the interaction and significance of coarse clinical labels and fine-grained biomarkers on OCT and fundus data. Their interactions (correlations) are made explicit. We then validate these interactions across the Experimental section with three (increasingly difficult) applications : a) Section 4.1 takes advantage of biomarkers and clinical data to make a binary classification for disease detection, b) Section 4.2 uses clinical labels for a pretext task to contrastively detect multi-label biomarkers, and c) Section 4.3 uses the established BCVA as labels for treatment prediction visit-by-visit.
> >
> > 4. We provide t-SNE embedding visualizations for the contrastive learning experiments in Fig. 6. This is valuable in emphasizing the correlations and significance of data modalities.
> >
> > **Clinical labels and biomarkers are associated with each eye. How is the relation across the two modalities developed for the datasets? Per my understanding, there should be a 'Patient id' with 'left' and 'right' eyes and corresponding clinical labels and biomarkers associated with each sample (eye).**
> >
> > A more complete understanding of the setup of the data can be understood from a discussion of the label csv files in Appendix D.2. In Figure 12, we see an example of a subset of label CSV file. From the first row, we observe that the path to every single image is specified with the format (Trial/Arm/Folder/Visit/Eye/Image Name). Within appendix section D.2 (Line 1102), there is a more detailed description of the path information. After this path information is the biomarkers and clinical labels associated with that specific image, which includes the patient ID. It should be noted that each ‘eye’ on an individual visit is associated with 49 images that are specified through the path information in the dataset. Within the path information, is a reference to whether the eye is the left or right. In the case of the PRIME and TREX clinical studies, oftentimes a single eye is treated and studied while the other eye is not utilized for treatment. As a result, most patients only have a single eye entered, with only 8 patients having both eyes in the dataset. For the small number of patients with both eyes entered, they are treated as individual entities and undergo different treatment studies.
> >
> > Every eye contributes:
> >
> > 1. An OCT Volume with 49 slices
> > 2. A single fundus photo
> > 3. Clinical information that comes from eye exams as well as measurements: (BCVA, CST, Patient, and Eye ID)
> > 4. 16 Biomarkers for every one of the 49 slices in the OCT Volume, these biomarkers are an interpretation of the conditions that exist within every slice.
> >
> > The relationship between clinical labels and biomarker labels are that clinical labels exist at the total eye level. In other words, clinical labels correspond to measurements that exist within the entire eye. These measurements give an overview of the condition of the eye, but they do not enable fine-grained analysis of structures that exist within the eye. Biomarker labels exist at the slice level.
> >
> > **Are the mentioned three modalities correspond to the same patient? This means there are three samples across three modalities for each patient**
> >
> > Yes. We reformatted Table 1 to effectively communicate the statistics of the dataset from a per patient/per visit perspective.
> >
> > **Data collected from a single center might encourage data bias.**
> >
> > We agree and acknowledge this in the Dataset Limitation section (Line 322). Moreover, we encourage the users of our dataset to test the capability of out of distribution experiments by utilizing other OCT datasets in tandem with ours. We provide references to these potential datasets from France, China, UK, collaboration between USA and Chine in Section B.4 that are openly accessible.
> >
> > Additionally, we use the Kermany dataset to learn a distribution of healthy images in order to gather statistics for the unlabeled images in the OLIVES dataset (Appendix C.7). Our dataset provides a wide range of research directions such as multi-modality, clinical information, and treatment information, but in terms of additional studies, it is entirely possible to leverage the data present in other datasets for the tasks that we provide.

---

> > > ### Author Response · Authors · 2022-08-24
> > > **Validity and significance of features from the results**
> > >
> > > We continue the rebuttal from the previous comment
> > >
> > > **In Table 3, many inputs used to train the model have shown a random accuracy for binary classification. This proves the insignificance of these features and contradicts the author's claims. Similar results are found in table 7 in the supplementary.**
> > >
> > > Please note that the three tasks in Section 4 are ordered by their difficulty, each of which depicts the value of the considered features. We describe the three tasks and quantify their results:
> > >
> > > 1. In Task 1 (Section 4.1 and Table 2), we make a disease classification based on OCT images. Most of the existing works on ophthalmic data (Fundus or OCT) generally use this setup. Existing works on multi-modal data, joint learning, and fusion can be leveraged for this purpose. Hence, baseline results are 70% while joint-learning based multi-modal fusion with biomarkers provide a 12% increase in Table 2. Hence, Biomarkers and clinical labels significantly contribute to these increased results and existing frameworks are able to leverage them.
> > >
> > > 2. In Task 2 (Section 4.2 and Table 3), we present two results:  a) detecting individual biomarkers from OCT scans, and b) detecting the presence of 5 biomarkers at the same time. These two related tasks aren’t inherently equivalent because of the nature of biomarkers. For example, DME, IRF, and FAVF have detection accuracies between 70% and 80%, while PAVF and IRHRF have accuracies between 50% - 70%. In Fig. 7 of the revised paper, we show OCT scans for each biomarker. While DME, IRF, and FAVF are readily apparent throughout the scans, PAVF and IRHRF manifest as small perturbations. DME, IRF, and FAVF detection fits closely to the setting of the natural image domain where, oftentimes, classification occurs on images that have the subject of interest at the forefront. This is not the case, however, with PAVF and IRHRF. An ideal algorithm would be able to perform well in both cases, but in reality a model is more likely to “overfit” to the features that are easy to distinguish and lose the capability to detect the much more difficult fine-grained features. Our results indicate that while performance on the fine-grained biomarkers PAVF and IRHRF are not as good compared to the others, there is a significant performance increase compared to the state of the art self-supervision strategies by incorporating clinical information into the contrastive learning strategy (52% →65% on PAVF) and (59% →64% on IRHRF). This represents a step forward in trying to overcome this difference between identification of both coarse and fine-grained features. The second part of the table shows multi-label AUROC, average sensitivity, and average specificity. The intent of these results is to show how well each algorithm does in the task of predicting all 5 of these biomarkers at the same time. We point to performance ranging from .7 to .8 on these metrics as evidence of the significance of these features within a biomarker detection task.
> > >
> > > 3. In Task 3 (Section 4.3 and Table 10), we provide treatment prediction on a visit-by-visit basis. Specifically, in Table 10, we add benchmarks using ResNet-50, DenseNet-121, EfficientNet and Vision Transformers to identify if the ocular health (as measured by BCVA) improves every visit. This is a hard problem since treatment affects different individuals differently. Manifestation of any disease and consequently its treatment in a person is a product of multiple factors including genes, environment, and lifestyle for each person [2]. We provide both an ML perspective as well as a medical perspective for this difficulty in Appendix C.3. We are unaware of large scale datasets that allow for this setup. For instance, the authors in [1] model disease progression on 3308 OCT scans by predicting if the disease manifestation becomes severe. Hence, they model it as a single classification problem. We present a similar task In Fig. 10, where we predict the final state of the biomarkers after treatment given the initial set of biomarkers for patients. The predicted results (green in Fig. 10) follows the ground-truth (red) with a large overlap, validating the features. However, because of the larger set of images as well as label modalities, we believe that OLIVES can spur further research on a wide variety of explicit time-series association models, for a visit-by-visit basis. The highest current results are 60% from EfficientNet in Table 10 and this is a testament to the difficulty of the task.
> > >
> > > [1] Rivail, Antoine, et al. "Modeling disease progression in retinal OCTs with longitudinal self-supervised learning." International Workshop on PRedictive Intelligence In MEdicine. Springer, Cham, 2019.
> > >
> > > [2] Straatsma, Bradley R. "Precision medicine and clinical ophthalmology." Indian Journal of Ophthalmology 66.10 (2018): 1389.

---

### Official Review · Reviewer_yk7T · 2022-07-27
**Well-written paper with potentially positive impact on medical research and treatments**

**Rating:** 7
**Confidence:** 1
**Clarity:** yes, the writing style seem good and …

**Strengths:**

The paper provides medical data from different modalities with potentially positive impact on medical research and treatments.

The authors train different ML models to analyze the ability of the presented data to detect the relevant diseases ( DR/DME ), as well as  predicting the effects of the successive treatment and the final occular state.

They explain the technical details of their experiments and their outcomes.

The paper & the pesented dataset seem good, grounded and valuable to me, but it is hard for me as someone without any medical background to evaluate the medical analysis and justifications made in this paper.

**Weaknesses:**

It is unclear to me, why they trained the vision models used to test the abilities of the dataset with a ResNet 18 backbone, which is pretty small and old compared to 2022 ML standards.



**Additional Feedback:**

It would be interesting to evaluate the dataset using newer, potentially better models e.g. by finetuning the Image encoder backbone of a CLIP model.

**Correctness:**

In general, it is hard for me as someone without any medical background to evaluate the medical analysis and justifications made in this paper.

**Documentation:**

Good.

**Ethics:**

The paper has a well-written "Limitations and Societal Impact" that reffers to other existing datasets that couldbe used as test sets from different populations.

**Relation To Prior Work:**

I cannot judge this, cause this medical field is new to me.

**Summary And Contributions:**

The paper provides medical data from different modalities with potentially positive impact on medical research and treatments.

The authors train different ML models to analyze the ability of the presented data to detect the relevant diseases ( DR/DME ), as well as  predicting the effects of the successive treatment and the final occular state.

They explain the technical details of their experiments and their outcomes.

---

> ### Author Response · Authors · 2022-08-24
> **Results on additional architectures**
>
> We thank the reviewer for their positive comments. We address the question of newer benchmarks by adding them in the appendix.
>
> **It is unclear to me, why they trained the vision models used to test the abilities of the dataset with a ResNet 18 backbone, which is pretty small and old compared to 2022 ML standards.**
>
> We add additional benchmarks on architectures including ResNet-50, Densenet-121, EfficientNet, and Vision Transformers in Table 10. We do this on time-series data on a hard problem: predicting the treatment effects on a visit-by-visit basis. From an ML perspective this is hard since there are only fine-grained domain shifts between visits due to time and treatment. From a medical perspective, this is hard since images themselves are sometimes insufficient to characterize the treatment effect and require other mult-modal data in clinical labels and biomarkers. Hence, this suggests the requirement of further research into treatment analysis preferably in a multi-modal setting. We expand on this in Section C.3.

---

### Official Review · Reviewer_E6sq · 2022-07-27
**OLIVES Dataset: Ophthalmic Labels for Investigating Visual Eye Semantics**

**Rating:** 5
**Confidence:** 2
**Correctness:** Yes.
**Clarity:** Yes.

**Strengths:**

This dataset contains 96 eyes and an average of 16 visits per patient, and 1268 fundas eye images. Figure 1 clearly illustrates the clinical practice described in section 1.

This dataset is collected over a long period of time. The long spanning of time series data allows future researchers to perform experiments on predictive models.

Good level of details on data collection and hyperparameters used.

The authors have discussed related work in different aspects. All mentioned research was properly referenced.

Compared with existing datasets, OLIVES contains a comprehensive set of modalities and is large enough in volume to be leveraged by ML algorithms. According to the paper, this is currently the largest and most diverse dataset of its kind.


**Weaknesses:**

The entire paper is hard to follow for reviewers who are not experts in biology because of the extensive use of abbreviations of biological terminologies. I understand that this paper is targeted toward an audience in the domain of biology/medicine. Still, to facilitate interdisciplinary research, it would be great if the authors could include in their appendix the corresponding full names of the abbreviations used in the paper.

I would suggest the authors reorganize section 4.1. Table 2 presents experiments with increasing balanced accuracy, but section 4.1 presents different tasks in different orders, which makes readers hard to follow. Would be great if the authors could indicate which ML model is used for which task in the tables.

Table 3 is unclear at first glance. It would be clearer if the authors could discuss the first three models in detail in the corresponding section. Also, it would be better to mention that table six is in the appendix. At first glance, I thought the authors forgot to present table six in the paper. Also, in the last section of section 4, there is no “figure c.3” in appendix “c.3”. The overall paper needs more careful review.

In the discussion section, the authors should consider elaborating more upon the ethical implications of this study.


**Additional Feedback:**

 N/A.

**Documentation:**

Yes.

**Ethics:**

See weakness.

**Relation To Prior Work:**

Yes.

**Summary And Contributions:**

This paper presents OLIVES, an OCT and near-IR fundus dataset that includes clinical labels, biomarker labels, disease labels, and time-series patient treatment information from associated clinical trials. The dataset contains the information of 96 eyes averaged over a period of at least two years, with each eye treated for an average of 66 weeks and 7 injections. Benchmark experiments, benchmark models, and baseline results are presented.

---

> ### Author Response · Authors · 2022-08-24
> **Improving readability from an interdisciplinary standpoint**
>
> We thank the Reviewer for their comments and address them individually below.
>
> **The entire paper is hard to follow for reviewers who are not experts in biology because of the extensive use of abbreviations of biological terminologies. I understand that this paper is targeted toward an audience in the domain of biology/medicine. Still, to facilitate interdisciplinary research, it would be great if the authors could include in their appendix the corresponding full names of the abbreviations used in the paper.**
>
> We thank the reviewer for their comments. We are very keen to have OLIVES dataset be accessible to interdisciplinary researchers. As such, we have made the following changes:
>
> 1. We have increased the ‘Challenging dataset for ML Research’ section in Introduction. Here, we motivate the challenges associated with multi-modal data from both a medical and ML sense using Fig.2 that was previously in the supplementary.
>
> 2. We move the clinical trial descriptions from Section 3.1 to the appendix in D.1 to not confuse the readers regarding prospective studies.
> 3. We define biomarkers and their relationship with diseases before describing their labeling in Section 3.1, Line 129.
>
> 4. We expand on the significance of the clinical labels, their meaning and ranges in Section 3.2. Specifically, clinical labels provide coarse measurement for the entire eye while biomarkers provide fine-grained labels on 3D longitudinal slices of the eye.
>
> 5. We add a new Subsection 3.4 that details the interaction and significance of coarse clinical labels and fine-grained biomarkers on OCT and fundus data. Their interactions (correlations) are made explicit. We then validate these interactions across the Experimental section with three (increasingly difficult) applications : a) Section 4.1 takes advantage of biomarkers and clinical data to make a binary classification for disease detection, b) Section 4.2 uses clinical labels for a pretext task to contrastively detect multi-label biomarkers, and c) Section 4.3 uses the established BCVA as labels for treatment prediction visit-by-visit.
>
> 6. We provide t-SNE embedding visualizations for the contrastive learning experiments in Fig. 6. This is valuable in emphasizing the correlations and significance of data modalities.
>
> 7. We add Table 6 that provides full names of all abbreviations used in the paper.
>
> 8. We provide two sets of labels: a) Full labels that have all the details from the clinical trials, and b) ML centric labels with only required data. This is shown in Figs. 12, 13 and expanded in Appendix D.2.
>
> If the reviewer has any additional suggestions to improve interdisciplinary readability, we are keen to hear and implement it.
>
> **I would suggest the authors reorganize section 4.1. Table 2 presents experiments with increasing balanced accuracy, but section 4.1 presents different tasks in different orders, which makes readers hard to follow. Would be great if the authors could indicate which ML model is used for which task in the tables.**
>
> Thank you. We have applied these changes to the paper as well as ML models.
>
> **Table 3 is unclear at first glance. It would be clearer if the authors could discuss the first three models in detail in the corresponding section. Also, it would be better to mention that table six is in the appendix. At first glance, I thought the authors forgot to present table six in the paper. Also, in the last section of section 4, there is no “figure c.3” in appendix “c.3”. The overall paper needs more careful review.**
>
> Thank you. We have gone through the paper and have corrected the references to tables, figures, and appendices numbers. We have explicitly referenced all tables and figures in the appropriate locations in the main paper. We have made changes to the text in Section 4.2 to reflect the compared contrastive learning method against our own.
>
> **In the discussion section, the authors should consider elaborating more upon the ethical implications of this study.**
>
> We add ethical concerns regarding the ML side of the research in the Limitations, Societal Impact and Ethical Concerns section in blue. It is reproduced below:
>
> *We believe that a combination of datasets taken over multiple geographical regions, times, and disease states is essential to construct generalizable and ethical ML models. ML models can potentially amplify existing inequalities within healthcare access [41]. For instance, the data in OLIVES is collected from December 2013 to April 2021, which implies the participants had the time and means to be part of these trials. This may not always be the case for disadvantaged groups. Hence, any benefit that machine learning could provide will be restricted to small subsets of society unless thought is put into preventing this disparity. Hence, a careful analysis of potential concerns is required to use OLIVES and any other dataset to enrich the functionality and adaptability of machine learning algorithms in everyday lives.*

---

### Official Review · Reviewer_4cgE · 2022-07-28
**OLIVES Dataset: Ophthalmic Labels for Investigating Visual Eye Semantics**

**Rating:** 7
**Confidence:** 3
**Correctness:** Yes

**Strengths:**

- Longitudinal data covering multiple modalities
- Facilities research in several different directions: disease detection and classification, treatment progression as well as understanding the relationship between multiple modalities.

**Weaknesses:**

- Authors explicitly state this, data is collected from one geographical location and could be biased.
- Sample size in terms of the number of patients.


**Additional Feedback:**

This sort of longitudinal datasets with multimodal data are very important and hard to acquire. The concerns I have with this dataset are the limitations in terms of the number of patients and the lack of geographic diversity.

**Clarity:**

The paper is well written. The machine learning tasks mentioned in the Section 4 could elaborate on the inference task.

**Documentation:**

Yes

**Ethics:**

No personally identifiable information is shared, so I don't think there are any ethical concerns.

**Relation To Prior Work:**

Yes

**Summary And Contributions:**

Authors presented a longitudinal multi-modal dataset comprising of 2D fundus image, 3D OCT scans, clinical labels and biomarkers collected from patients undergoing treatment for Diabetic Retinopathy or Diabetic Macular Edema. Authors have also presented baseline results for tasks such as DR/DME detection, disease classification, biomarker detection and clinical outcome prediction.

---

> ### Author Response · Authors · 2022-08-24
> **Addressing dataset diversity**
>
> We thank the reviewer for their positive comments. We agree with the reviewer regarding the sample size as well as the possible bias and we have conceptually and experimentally addressed this in the revision.
>
> **Authors explicitly state this, data is collected from one geographical location and could be biased**
>
> As the reviewer notes, we address this and provide links to other open-access datasets (Line 324 and Appendix B.4). The OLIVES dataset is the outcome of trials that ran from 2013 to 2021. Please note that this is not uncommon for datasets collected with time-series labels. The dataset in [1] is also the outcome of 7 years worth of trials. Hence, these datasets require large time and personnel commitments to obtain and curate. We firmly believe that a combination of datasets taken over multiple geographical regions, times, and disease states are essential to construct generalizable ML models. To this end, we incorporated the Kermany dataset [13 in paper] in our experiments in Appendix C.7, Table 13. In Table 13, we show that applying natural-image based SOTA contrastive learning algorithms on ophthalmic data is inferior to contrastive learning algorithms applied on OLIVES+Kermany datasets. The diversity within the two datasets induces higher results. We state this belief of generalizing on top of multiple clinical studies in the societal impact section.
>
> **Sample size in terms of the number of patients.**
>
> In Table 5, we compare against other datasets that provide time-series data. Because of the large time commitments, there are not many. The dataset from [1] has more eye data than compared to OLIVES. However, the number of scans is 3308 which is far less than our 78,185 scans.  Hence, [1] models their algorithm as a single classification challenge of predicting severity of disease. Because of the larger set of images as well as label modalities, OLIVES allows for visit by visit analysis (Section 4.3). Other clinical study papers that provide time series data include [2] which is performed over 193 eyes. However, this is a clinical study without a curated ML dataset. PRIME clinical study has 40 eyes and TREX DME has 150 eyes, thus making their combination similar to [2]. However, after accounting for quality, multi-modality, and visit recurrences, we provide a highly curated dataset of 96 eyes that is ML-centric and open source.
>
> [1] Rivail, Antoine, et al. "Modeling disease progression in retinal OCTs with longitudinal self-supervised learning." International Workshop on PRedictive Intelligence In MEdicine. Springer, Cham, 2019.
>
> [2] Vogl, Wolf-Dieter, et al. "Analyzing and predicting visual acuity outcomes of anti-VEGF therapy by a longitudinal mixed effects model of imaging and clinical data." Investigative Ophthalmology & Visual Science 58.10 (2017): 4173-4181.

---

> > ### Comment · Reviewer_4cgE · 2022-09-03
> > **Response to the authors**
> >
> > I would like to thank the authors for their detailed response. I believe their experiments can only partly address the concerns with diversity and coverage. However, I would like to champion this paper as I strongly believe collection of this kind of longitudinal data is very important and quite hard to get the execution right.

---

### Official Review · Reviewer_YjDo · 2022-07-28
**New biomarkers and classification benchmarks for ophthalmology data**

**Rating:** 4
**Confidence:** 2
**Clarity:** The paper seems to be well written, b…

**Strengths:**

- The authors identified good ML tasks relevant to Ophthalmology patient care, and how they can be incorporated into patient care
- The authors showed good summaries of the labels / biomarkers created for the data.

**Weaknesses:**

- It would be nice if the authors can elaborate more on the clinical significance of the biomarkers, why were these markers chosen, and what are the value ranges of the biomarkers and their implications.
- It is unclear about the grader’s qualification, and how the biomarkers are acquired (is there only one grader for each scan or are there multiple graders for each scan)
- There are standard deviations for the balanced accuracy. Why aren’t they included for other metrics?
- It is unclear if there are any domain shift between the data collected from two different studies. They study different conditions, which are also the labels of the classification task. Any domain shift between the datasets would compromise the classification result. It is also unclear why these two datasets are selected, and what are the clinical impact of such a combined dataset.
- Table 1 is confusing, it is unclear if there is any relationships between the merged rows in image modalities and label modalities.
- Time series data in the dataset correspond to visits, which have an average of 16 data points, and may have different frequencies and intervals. It us unclear how these data contribute to the classification of patients and there are no benchmarks included in the paper based on the time series data.

**Additional Feedback:**

N/A

**Correctness:**

The dataset seems to be properly constructed and the experimentation seems to be reasonable

**Documentation:**

Data, label, and source code are available for review

**Ethics:**

No ethical concerns

**Relation To Prior Work:**

The authors described how the work relates to prior work and compared it to existing datasets

**Summary And Contributions:**

The authors combined existing ophthalmology datasets, and introduced additional biomarkers / labels. The authors identified ML tasks that are relevant to patient care, and benchmarked classification performance on image data, biomarkers, and multimodal inputs.

---

> ### Author Response · Authors · 2022-08-24
> **Changes made to effectively bring out the significance of the labels**
>
> We thank the Reviewer for their comments and address them individually below.
>
> **It would be nice if the authors can elaborate more on the clinical significance of the biomarkers, why were these markers chosen, and what are the value ranges of the biomarkers and their implications.**
>
> We thank the reviewer for this question and have addressed it in the revised paper in Section 3.1. The biomarkers themselves are chosen because of their structural properties that might indicate disease states (We add this in Line 146). However, they are not causal. We emphasize this by adding the following paragraph from Line 129 onwards:
>
> *Biomarkers are quantifiable characteristics of biological processes in the eye. In this paper, the biological processes are diseases and biomarkers indicate the presence or absence of such diseases. Under limited circumstances, the authors in [1] suggest that biomarkers can be surrogate endpoints in clinical trials. However, they caution against doing so unless the underlying clinical trial is specifically meant for the study. In both the PRIME and TREX DME studies, biomarkers are retrospectively labeled. As such, biomarkers may indicate the presence of diseases, but are not causal to these diseases. Hence, biomarkers are different from visual causal features from [23] or causal question-based analysis in [24] or causal factor analysis in [25].*
>
> Section 3.1 along with Figure 3 shows a histogram distribution of the biomarkers within the dataset and states the implication behind this distribution. In terms of value ranges, biomarkers are one-hot labels (Line 150). They are binary labels with 1 indicating existence of a particular biomarker and 0 indicating otherwise.
>
> Similarly, we expand on the significance of the clinical labels, their meaning and ranges in Section 3.2. Specifically, we reproduce the changes from Lines 160 onwards:
>
> *ETDRS best-corrected visual acuity (BCVA) is a visual function assessment performed by certified examiners where a standard vision chart is placed 4-meters away from the patient. The patient is instructed to read the chart from left to right from top to bottom until the subject completes 6 rows of letters or the subject is unable to read any more letters. The examiner marks how many letters were correctly identified by the patient. Central subfield thickness (CST) is the average macular thickness in the central 1-mm radius of the ETDRS grid. Both BCVA and CST are coarse measurements over the eye as opposed to Biomarkers that exist for fine-grained longitudinal slices of the eye. BCVA can range from 0-100 and CST from 100-1300.*
>
> Hence, the interaction and significance of coarse clinical labels and fine-grained biomarkers on OCT and fundus data is established in Sections 3.1 and 3.2 and their interactions (correlations) is made explicit in a new Section 3.4. We then validate these interactions across the Experimental section with three (increasingly difficult) applications : a) Section 4.1 takes advantage of biomarkers and clinical data to make a binary classification for disease detection, b) Section 4.2 uses clinical labels for a pretext task to contrastively detect multi-label biomarkers, and c) Section 4.3 uses the established BCVA as labels for treatment prediction visit-by-visit.
>
> We believe that the additions have brought out the significance of the labels and ultimately the paper and thank the Reviewer for this and the further comments.
>
> **Table 1 is confusing, it is unclear if there is any relationships between the merged rows in image modalities and label modalities.**
>
> We reformatted Table 1 to effectively communicate the statistics of the dataset from a per patient/per visit perspective.
>
> **It is unclear about the grader’s qualification, and how the biomarkers are acquired (is there only one grader for each scan or are there multiple graders for each scan)**
>
> In appendix sections B.5.1 and B.5.2, the exact labeling process for biomarker and clinical labels is discussed in detail. In appendix section B.3, the dataset logistics and the qualifications of the graders are mentioned. Specifically, Line 719 onwards:
>
> *The biomarkers are identified by Charles C. Wykoff with an ophthalmology experience of sixteen years and the labeling is performed by Stephanie Trejo Corona with a grading experience of one year.*
>
> Also, for difficult cases, the standard practice for medical grading was followed. Specifically, in Line 149:
>
> *Open adjudication was done with an experienced retina specialist for difficult cases.*
>
> **There are standard deviations for the balanced accuracy. Why aren’t they included for other metrics?**
>
> Standard deviations for Table 3 is present in Table 9 in the revised paper. Additionally, all Tables 8, 9, 10, 11, and 12 and Fig. 10 in the supplementary have standard deviations. The reason this was moved to the appendix is due to space constraints. We specifically indicate this in the main paper (caption of Table 3).

---

> > ### Author Response · Authors · 2022-08-24
> > **Domain Shift and Time-Series experiments**
> >
> > We continue the rebuttal in this comment.
> >
> > **It is unclear if there are any domain shift between the data collected from two different studies. They study different conditions, which are also the labels of the classification task. Any domain shift between the datasets would compromise the classification result. It is also unclear why these two datasets are selected, and what are the clinical impact of such a combined dataset.**
> >
> > We thank the reviewer for these comments. We performed and included a number of additional experiments to address this question. We answer this question qualitatively and quantitatively:
> >
> > 1. Qualitative:  A full discussion of each of the two trials are presented in appendix D.1. PRIME studies DR while TREX studies DME diseases. The eyes in TREX are studied over three years and have more severe conditions than those present in PRIME. For this reason, combining the two trials allows for a more complete distribution in terms of the structural specifications of the eye. This is opposed to having a single clinical trial where most of the images are healthy or most of the images are diseased. Moreover, combining the studies allows for studying differentiating factors between DR/DME instead of between healthy vs single-diseased. Logistically, both trials are conducted at the exact same clinic within a similar population group using the same imaging setup. Hence, any domain difference that exists between trials is due to the disease manifestation, treatment over the course of three years, and multi-modal data. We expand on this in Fig. 2 and Line 60 in the Introduction.
> >
> > 2. Quantitative:  In Table 11, we perform additional experiments where we perform inta-trial vs inter-trial experiments. Intra-trial refers to within PRIME and within TREX experiments - train and test within respective trials. Inter-trial refers to training and testing on different trials. The best results are obtained when training and testing on TREX - this is because of diversity in TREX data due to the larger clinical trial window of 3 years. For this reason, combining the two trials provides the best results as shown in Table 3 (which are combined). Interestingly, the inter-trial results when training on TREX and testing on PRIME is higher than intra-trial training and testing on PRIME.
> >
> > However, domain shift can also occur due to treatments. Both time and treatment changes the manifestation of diseases, thereby domains. We perform two additional experiments to analyze and account for this shift:
> >
> > 1. We experiment by training and testing before and after treatments (first and last visits) and show that ML models are susceptible to this domain shift. This is discussed in Appendix C.5 and Table 12.
> >
> > 2. We perform domain adaptation experiments where we take a part of the final visit data and append it to the first visit data (10%, 20%, and 30%) and perform DR/DME detection. The balanced accuracy with only OCT images shows an increase of 35% in Fig. 11. However, adding biomarkers to this setup adds an additional average of 10% to the results, showcasing the value of multi-modal data.
> >
> > We discuss this in the Discussion and Conclusion Section - Lines 303 onwards.
> >
> > **Time series data in the dataset correspond to visits, which have an average of 16 data points, and may have different frequencies and intervals. It is unclear how these data contribute to the classification of patients and there are no benchmarks included in the paper based on the time series data.**
> >
> > In the Introduction, we identify tasks related to Ophthalmology. One of these is treatment analysis. We pursue this in Section 4.3. However, due to space constraints, the results are presented in Appendix C.3. Specifically, in Table 10, we show benchmarks using ResNet, DenseNet, EfficientNet and ViTs to predict the treatment progression on a visit-by-visit basis. We identify if the ocular health (as measured by BCVA) improves every visit. Please note that there are multiple variations of the setup where BCVA can be replaced with any other multi-modal label. This is a hard problem and we provide both an ML perspective as well as a medical perspective for it in Appendix C.3. In Fig 10, we predict the final state of the biomarkers given the treatment and the initial set of biomarkers for patients. Please note that treatment analysis is notoriously hard and we are unaware of large scale datasets that allow for it. For instance, the authors in [1] model disease progression on 3308 OCT scans by predicting if the disease manifestation becomes severe. Hence, they model it as a single classification problem. Because of the larger set of images as well as label modalities, OLIVES allows for visit by visit analysis.
> >
> > [1] Rivail, Antoine, et al. "Modeling disease progression in retinal OCTs with longitudinal self-supervised learning." International Workshop on PRedictive Intelligence In MEdicine. Springer, Cham, 2019.

---

### Author Response · Authors · 2022-08-24
**Summary of the revised submission**

We thank the Program chairs, Area chairs and Reviewers for their work. We thank the reviewers for their feedback. We have addressed each of the reviewer questions individually. All changes in the paper are in blue. The new additions to the paper are listed below:

1. We reformatted Table 1 to effectively communicate the statistics of the dataset from a per patient/per visit perspective.

2. We add a new Interaction of Data Modalities subsection (Section 3.4) in Section 3. This section provides a high-level description about the data modalities to motivate their applicability in the benchmarking experiments of Section 4. Specifically, we provide high level descriptions of biomarkers (as fine-grained labels) and clinical labels (as coarse labels) in Sections 3.1 and 3.2 respectively.

3. Furthering this analysis, we provide t-SNE embedding visualizations for the contrastive learning experiments in Fig. 6. This is valuable in emphasizing the correlations and significance of data modalities.

4. We add an additional paragraph in the Discussion section regarding the domain shift and domain adaptation in OLIVES datasets. The experiments and results for this discussion are present in the supplementary (Appendix C.5 and C.6)

5. We add additional architectures including ResNet-50, Densenet-121, EfficientNet, and Vision Transformers in Appendix C.3, Table 10.

6. We add an additional experiment in Appendix C.7, Table 13 where we combine our dataset with existing Kermany dataset to showcase results.

We also moved a few sections to and from the supplementary materials. These are listed below:

1. We have expanded the Challenges in ML subsection within the Introduction by moving Fig.2 into the main paper instead of the Supplementary.

2. We move the PRIME and TREX DME clinical procedure details to Appendix D.1.

3. The Final Ocular state prediction is moved to Supplementary materials in Appendix C.3.

We are happy to address any additional questions that the reviewers have.

---

### Meta-Review · Area_Chair_NZAt · 2022-09-07

**Recommendation:** Accept
**Confidence:** 3

**Metareview:**

The reviewers struggled to find a consensus for this paper. Concerns about applicability of the dataset due to domain shift, issues with the data collection, size of the dataset, and clarity of the paper were raised. At the same time, I believe that despite its size, the value of longitudinal data for diagnostics is extremely valuable to the community. And the authors have made efforts to improve the readability, therefore I recommend accept.

---

### Decision · Program_Chairs · 2022-09-16

Accept